# Enhanced passive surveillance dengue infection among febrile children: Prevalence, co-infections and associated factors in Cameroon

**Celine Nguefeu Nkenfou** [1,2]*, **Nadine Fainguem**[1], **Félicitée Dongmo-Nguefack**[3,4], **Laeticia Grace Yatchou**[1], **Joel Josephine Kadji Kameni**[1], **Elise Lobe Elong**[1], **Amidou Samie**[5], **William Estrin**[6], **Paul Ndombo Koki**[3,4], **Alexis Ndjolo**[1,4]

1 "Chantal Biya" International Reference Centre for Research on HIV and AIDS Prevention and Management (CIRCB), Yaoundé, Cameroon, 2 University of Yaoundé I, Higher Teachers' Training College, Department of Biology, Yaoundé, Cameroon, 3 Mother and Child Centre of the Chantal Biya Foundation, Pediatric Service, Yaoundé, Cameroon, 4 University of Yaoundé I, Faculty of Medicine and Biomedical Sciences, Yaoundé, Cameroon, 5 Department of Microbiology, University of Venda, Venda, South Africa, 6 California Pacific Medical Center, San Francisco, CA, United States of America

* nkenfou@yahoo.com

**Data Availability Statement:** All relevant data are within the manuscript and its Supporting Information files.

## Abstract

Dengue virus (DENV) causes a spectrum of diseases ranging from asymptomatic, mild febrile to a life-threatening illness: dengue hemorrhagic fever. The main clinical symptom of dengue is fever, similar to that of malaria. The prevalence of dengue virus infection, alone or in association with other endemic infectious diseases in children in Cameroon is unknown. The aim of this study was to determine the prevalence of dengue, malaria and HIV in children presenting with fever and associated risk factors.

Dengue overall prevalence was 20.2%, Malaria cases were 52.7% and HIV cases represented 12.6%. The prevalence of dengue-HIV co-infection was 6.0% and that of Malaria-dengue co-infection was 19.5%. Triple infection prevalence was 4.3%. Dengue virus infection is present in children and HIV-Dengue or Dengue- Malaria co-infections are common. Dengue peak prevalence was between August and October. Sex and age were not associated with dengue and dengue co-infections. However, malaria as well as HIV were significantly associated with dengue (P = 0.001 and 0.028 respectively). The diagnosis of dengue and Malaria should be carried out routinely for better management of fever.

## Author summary

Fever is a symptom common to several infectious diseases such as malaria, yellow fever, dengue fever and typhoid which are all present in Cameroon. In most cases, fever in Cameroon is assumed to be due to malaria and treated as such. This study present the prevalence of dengue fever among febrile children. Dengue fever as well as malaria is common among Cameroonian children. Since Cameroon is endemic for malaria, fever may

**Funding:** The authors received no specific funding for this work.

**Competing interests:** The authors have declared that no competing interests exist.

indicate either malaria or dengue. Clinicians should be aware of the presence of dengue in Cameroon. Knowing the risk factors, parents are encouraged to use the bed net to protect their children and clean stagnant water from their environment. Children could be co-infected by dengue, malaria as well as HIV. Management of patients with dengue fever and HIV infection should be studied, especially in infants where both viral diseases are more severe. Close monitoring may prevent complications among HIV-infected individuals who contract dengue fever.

## Introduction

Dengue is currently regarded as the most prevalent mosquito-borne viral disease worldwide [1,2] and is described as a "neglected" tropical disease by the World Health Organization [3]. Over 50% of the world's population live in dengue endemic countries [3–5]. Malaria is the most important mosquito-borne parasitic disease. Both infections share similar clinical symptoms [2]. The true impact of dengue in most African countries is unknown due to inadequate surveillance, misdiagnosis or lack of diagnosis, and low level of reporting or no reporting at all [6]. In Cameroon, dengue fever is not screened routinely, neither in children nor in adults. There is a licensed dengue vaccine available in many countries, but safety concerns and screening requirements limit its widespread use and impact. Only accurate diagnosis and appropriate clinical management can reduce the morbidity and mortality associated to dengue fever.

Few dengue studies have been conducted in Africa in general and Cameroon in particular. The first cases of dengue were found among 2 expatriates in Cameroon (German) in 2002, with serotype 1 (DENV-1) [7]. Another study showed that the seroprevalence of dengue in Cameroon was 35.9% in 2004 [8]. A study of prevalence and distribution of Flaviviridae, Togaviridae, and Bunyaviridae arboviral infections, done on adults in rural Cameroon, showed that 12.5% of cases were due to dengue type 2 virus (DENV-2) [9]. A cross sectional survey using a random cluster sampling method conducted in 2009 reported dengue cases in Yaoundé, Douala and Garoua, in adults, with a seroprevalence higher in Douala (60.8%) probably because of the climate (hot and humid) that promote the development of the transmitting vector, *Aedes* [10–11], compared to 23.3% in Garoua (hot and dry) and 9%, in Yaoundé (temperate). No case of dengue hemorrhagic fever (DHF) has been reported in Cameroon.

Cameroon is a tropical country where HIV, malaria and dengue co-exist as important infectious diseases. Dengue fever has been documented in Cameroon with two serotypes of viruses (DENV-1 and DENV-2). However, few data exist on dengue fever in children, on dengue-malaria and on HIV-dengue co-infection. The natural history and clinical manifestations of co-infection with HIV and dengue has not been established.

Therefore, the objective of this study was to determine the prevalence of dengue fever, dengue-malaria and dengue-HIV co-infections and associated risk factors in a pediatric population presenting with fever in Yaounde, Cameroon.

## Material and methods

### Ethics statement

The study received the approval of the National Ethical Committee, authorisation n° 2013/03/086/L/CNERSH/SP) of Cameroon. Infants were enrolled only when their parent or guardian gave written proxy-consent. Older children, aged above 5 gave their assent before enrollment.

Parents and Participants were ensured of confidentiality, by attribution of codified identification numbers.

We enrolled children and infants consulting at the Centre Mère Enfant–CME in Yaoundé for fever. Children aged 6 months to 15 years, presenting with fever (axillary temperature above 37.5˚C) were enrolled in the study. They were screened clinically and only those presenting with fever without known reasons (abscess, urinary infections, pulmonary infections. . .) were retained in the study.

A structured questionnaire (see S1 Study Questionnaire) was used to collect some relevant clinical and socio-demographic data. Three milliliters of blood were collected in an EDTA-tube from each child presenting with fever complaints. This study was carried out during the months of April to October 2015; which are rainy months with maximum transmission rate of both dengue and malaria, best for the breeding of their vectors (Anopheles and *Aedes*). The following tests were done:

Malaria was diagnosed using gold standard method (microscopic examination of giemsa stained thick and thin blood smears). Dengue was diagnosed using the SD Bioline NS1 Ag+Ab combo rapid test according to the manufacturer's instructions (Standard Diagnostics, Inc) and with ELISA CAPTURE DENGUE IgM/IgG. In order to confirm current dengue infection, in case of NS1 positive test, primer specific amplification was carried out as previously described [12–13]. HIV was diagnosed using rapid tests following the national algorithm for children aged more than 18 months (SD Bioline HIV-1/2 3.0 for the first test and Ora Quick Rapid HIV-1 Antibody Test for the second test on all samples). For younger children (less than 18 months) they were diagnosed using Roche amplicor DNA PCR version 1.5 according to the instructions from the manufacturer and as described by Nkenfou [14]. Hematological tests like CD4 count and full blood count were performed. Urine analysis was done using urine stick test as described by the manufacturer. Infants presented with positive urinary test were excluded from the study.

## Definitions of cases

Current or recent dengue infection was defined on the basis of a positive result for NS1 Ag and / or IgM. Past infection was defined on a basis of a positive result of IgG alone.

## Statistical analysis

Analyses of data obtained from questionnaires were carried out using Microsoft Excel 2010 and the Statistical Package for Social Sciences (SPSS) program (PASW Statistics 20). The Chi square test was used to determine the significance between different variables and the level of significance was 5%.

## Results

### 1) Socio demographic description of the study population

A total number of 349 children fulfilled the selection criteria (fever with unknown clinical cause). They were 53.3% boys and 46.7% girls, aged 6 months to 180 months with a mean age of 3.2 years. These data are presented in Table 1. Most of the children (67.9%) slept under a protective bed net and 46.9% were living in an environment with stagnant water.

### 2) Clinical characterization and prevalence of infections in the study population

All children enrolled presented an axillary temperature higher than 37.5˚C. The highest temperature observed was 41˚C, the lowest temperature observed was 37.8˚C and the average

**Table 1. Description of the study population.**

|  | Total | % |
|---|---|---|
| **Sex** |  |  |
| Male | 163 | 46.7% |
| Female | 186 | 53.3% |
| **Age (years)** |  |  |
| ‹2 | 168 | 48.1% |
| 2–5 | 101 | 28.9% |
| ›5 | 80 | 22.9% |
| **Usage of Impregnated bed nets** |  |  |
| Yes | 237 | 67.9% |
| No | 112 | 32.09% |
| **Presence of stagnant water** |  |  |
| Yes | 122 | 31.2% |
| No | 227 | 46.9% |

temperature was 38.9˚C. A total of 44 children (12.6%) were vomiting, 25 (7.2%) had diarrhea and 40 (11.5%) had fatigue. Out of 349 enrolled children,183 were infected by plasmodium (52.7% (183/349)), among which 54 were girls and 55 were boys (see Table 2). There were 183 cases of *P. falciparum* and 74 cases of *P. vivax*. All the *P. vivax* were in co-infection with *P. falciparum*.

Forty-four out of the 349 enrolled children were HIV positive (12.6%) (see Table 2). Co-infection with malaria was found among 15 participants while HIV co-infection with Dengue was found among 21 children.

Children presenting dengue-positive test for NS1 Ag, IgM, IgG and IgM or IgG Abs Rapid Diagnostic Test (RDT) were 0.2% (1/349), 20.2% (70/349), 26.8% (93/349) and 29.1% (101/349) respectively (see Table 2).

Average CD4 count was 969 cells/mm$^3$, normal for children aged 1–15 years. Children having the lower CD4 count (240–400 CD4 cells/mm3) were more likely to be positve for malaria

**Table 2. Overall prevalence or seroprevalence of different infections, co-infections and symptoms among the study participants.**

| Infection or symptoms | Frequency | Percent |
|---|---|---|
| **Malaria (any infection with *Plasmodium*)** | 183 | 52.7 |
| ***Plasmodium falciparum*** | 183 | 52.7 |
| ***Plasmodium vivax*** | 74 | 21.3 |
| **HIV** | 44 | 12.6 |
| **Dengue (Overall cases)** | 101 | 29.1 |
| **IgG dengue** | 93 | 26.8 |
| **IgM dengue** | 70 | 20.2 |
| **Vomiting*** | 44 | 12.6 |
| **Diarrhea*** | 25 | 7.2 |
| **Fatigue*** | 40 | 11.5 |
| **Current dengue + malaria + HIV**** | 15 | 4.3 |
| **Current dengue + malaria**** | 68 | 19.5 |
| **Current dengue + HIV**** | 21 | 6.0 |

* These symptoms are present together with fever.

** Here we consider acute dengue not past exposure.

(20% compared to 10.8%; X2 = 0.71, p = 0.4); for HIV (40% compared to 0%; X2 = 31.08; p<0.001) and for dengue(20% compared to 5.3%; X2 = 2.96, p = 0.085).

The rate of current dengue-malaria co-infection was 19.5% (68/349) and current dengue-HIV co-infection was 6% (21/349).

Fifteen children presented positive tests for active dengue, malaria and HIV all together, representing 4.3% of the study population (see Table 2).

## 3) Risk factors associated with dengue and dengue co-infections in the study population

Infants sleeping under impregnated bed nets were less at risk of been infected by dengue virus (OR 2.1, 95%CI 0.12–0.39, P< 0.001) and *Plasmodium* (OR 2.55, 95%CI: 1.4–4, p<0.0001). Children living in stagnant water free environment were less at risk of being infected by the two pathogens (OR 8.85, 95%CI 4.6–16.7, p<0.001 and OR 2.7, 95%CI 1.7–44, P< 0.001 respectively for dengue and malaria). The peak transmission occurred during the months of August to October, corresponding to the shorter raining season in Yaoundé-Cameroon see Fig 1.

Sex and age were not associated with dengue or any dengue co-infections studied; see Tables 3 and 4 respectively.

Although no statistical significance was found between age and dengue Ab, the prevalence (IgM) and seroprevalence (IgG) increased steadily with age as shown in Fig 2. This may be reflective of cumulative exposure, underlining the fact that dengue transmission is high and endemic rather than epidemic. As well, as the child grows older, he become more independent in his movements and can pull away the bednet.

Table 4 shows a significant association between dengue, malaria and HIV (p <0.001). Current dengue infection was associated with *Plasmodium falciparum* as well as *Plasmodium vivax* both with p< 0.001 and with HIV with p = 0.014. Similarly, the association between co-infection of malaria as well as HIV with current dengue infections was statistically significant indicating an endemic co-existence and potential association between these infections. The

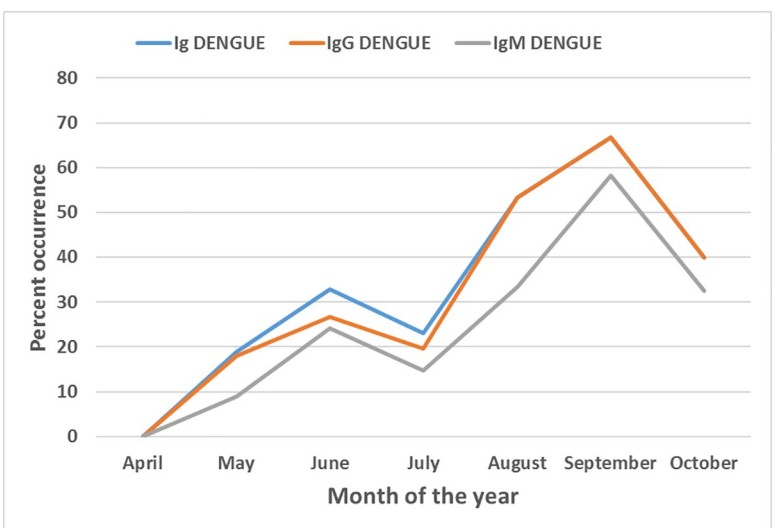

**Fig 1. Seasonal distribution of dengue virus infection among children with fever attending hospitals in Yaounde.**
Ig represents all cases of dengue infections (past and current infections). IgG represents cases of previous exposure to dengue. IgM represents current cases of dengue.

**Table 3. Sex distribution of the different infections (active and past) and symptoms in the study population.**

| | Sex | | Total | | |
|---|---|---|---|---|---|
| | **Female** | **Male** | **Total** | **X2, p value** | **OR; 95%CI** |
| **Ig dengue** | 54 (33.1%) | 47 (25.5%) | 101 | 2.410; 0.121 | 0.692; 0.435–1.102 |
| **IgG dengue** | 51 (31.3%) | 42 (22.8%) | 93 | 3.155; 0.076 | 0.650; 0.403–1.047 |
| **IgM dengue** | 33 (20.2%) | 37 (20.1%) | 70 | 0.001; 0.975 | 0.992; 0.586–1.677 |
| *P. falciparum* | 91 (56.5%) | 92 (49.5%) | 183 | 1.725; 0.189 | 0.753; 0.493–1.150 |
| *P. vivax* | 37 (23%) | 37 (19.9%) | 74 | 0.491; 0.484 | 0.832; 0.498–1.392 |
| **HIV** | 21 (12.9%) | 23 (12.4%) | 44 | 0.021; 0.884 | 0.954; 0.507–1.797 |
| **Vomiting** | 20 (12.3%) | 24 (12.9%) | 44 | 0.032; 0.859 | 1.059; 0.562–1.998 |
| **Diarrhea** | 10 (6.1%) | 15 (8.1%) | 25 | 0.486; 0.486 | 1.342; 0.586–3.076 |
| **Fatigue** | 15 (9.2%) | 25 (13.4%) | 40 | 1.538; 0.215 | 1.532; 0.778–3.018 |
| **Dengue + HIV** | 8 (4.9%) | 13 (7%) | 21 (6%) | 0.665; 0.415 | 1.456; 0.588–3.606 |
| **Dengue + Malaria** | 38 (23.3%) | 30 (16.1%) | 68 (19.5%) | 2.858; 0.091 | 0.633; 0.371–1.078 |
| **Dengue + Malaria + HIV** | 5 (3.1%) | 10 (5.4%) | 15 (4.3%) | 1.126; 0.289 | 1.795; 0.601–5.366 |
| **Total** | 163 | 184 | 347 | | |

Note: For the ODD ratios, Male is considered as the reference. In the case of co-infections, only acute dengue was considered.

possibility of a weakened immune system among pediatric patients could explain such association. Among the symptoms recorded during the consultation, diarrhea was associated with dengue, mainly the current infections (p = 0.036); see Table 5.

## Discussion

The sero-prevalence and distribution of dengue has been poorly documented in Cameroon and more so in children. A total of six studies reported on dengue in Cameroon, with two including children. We found a dengue prevalence (NS1, anti-dengue IgM antibodies detection) of 20.2% and a seroprevalence of 26.8% for anti-dengue IgG antibodies. The result obtained in the present study is higher than the 9.8% previously reported by Demanou et al in

**Table 4. Association between dengue (current and past), malaria and HIV.**

| | Dengue | | Total | | |
|---|---|---|---|---|---|
| | **No** | **Yes** | | **X2; p** | **OR, 95% CI** |
| | **Ig dengue** | | | | |
| **Malaria-PF** | 115 (47.1%) | 68 (67.3%) | 183 (53%) | 11.697; 0.001 | 2.311; 1.422–3.758 |
| **Malaria-PV** | 42 (17.2%) | 32 (31.7%) | 74 (21.4%) | 8.877; 0.003 | 2.231; 1.306–3.808 |
| **HIV** | 25 (10.2%) | 19 (18.8%) | 44 (12.7%) | 4.838; 0.028 | 2.048; 1.071–3.916 |
| | **IgG dengue** | | | | |
| **Malaria-PF** | 116 (46%) | 67 (72%) | 183 (53%) | 18.453; 0.0001 | 3.021; 1.803–5.062 |
| **Malaria-PV** | 42 (16.7%) | 32 (34.4%) | 74 (21.4%) | 12.691; 0.0001 | 2.623; 1.527–4.506 |
| **HIV** | 27 (10.6%) | 17 (18.3%) | 44 (12.7%) | 3.598; 0.058 | 1.881; 0.972–3.639 |
| | **IgM dengue** | | | | |
| **Malaria-PF** | 132 (48%) | 51 (72.9%) | 183 (53%) | 13.842; 0.0001 | 2.908; 1.632–5.180 |
| **Malaria-PV** | 47 (17.1%) | 27 (38.6%) | 74 (21.4%) | 15.281; 0.0001 | 3.046; 1.715–5.411 |
| **HIV** | 29 (10.5%) | 15 (21.4%) | 44 (12.7%) | 6.061; 0.014 | 2.332; 1.172–4.642 |
| **Total** | 244 | 101 | 345 | | |

Note: The absence of the other infection was considered as the reference.

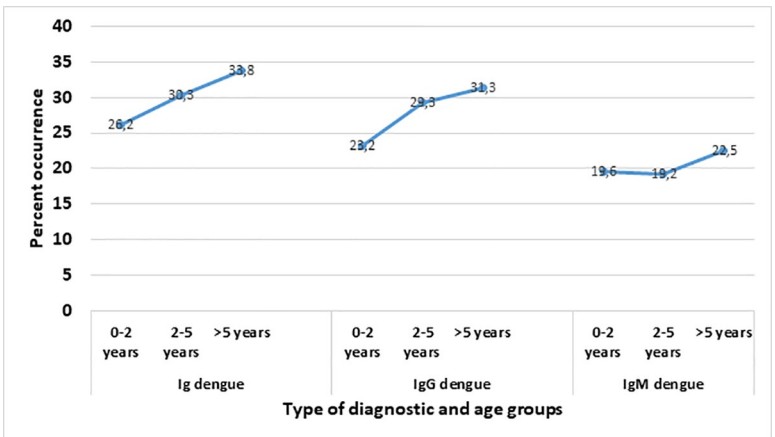

**Fig 2. Diagram showing the variation of the prevalence of dengue in relation to age group.** Ig represents all cases of dengue infections (past and current infections). IgG represents cases of previous exposure to dengue. IgM represents current cases of dengue.

Yaoundé among adults [11]. This was a cross sectional survey conducted in three main cities in Cameroon (Douala, Garoua and Yaounde), using a random cluster sampling design to document risk factors associated with anti-dengue virus Immunoglobulin G seropositivity in humans The higher rate of dengue in the present study may be explained by the higher susceptibility to infection in children and cases of mother to child transmission of dengue virus [15–16].

The primary clinical manifestation of dengue is fever, similar to that of malaria which is endemic in Cameroon [17]. Our findings showed a prevalence of 52.7% of malaria cases alone and 19.5% for malaria-dengue co-infections in our study population. The first documented cases of dengue-malaria co-infection among children in Cameroon was published in 2018 [18] with a prevalence of 12.5%. This percentage is lower than that obtained in our study. We had a proportion of 19.5% compared to 12.5% in the study of Monamele *et al* in 2018. The reason may be that the methodology used in the two studies for malaria diagnosis was different, we used microscopy, still considered as gold standard, while in the previous study the rapid diagnostic test was used. As well their population size of 33 malaria cases was smaller than our which was 183 cases.

Acute dengue-malaria co-infection could present more severe clinical symptoms than a single infection [19]. These findings show that there is need for more attention in case of fever to carry out a laboratory test for malaria or dengue before treatment.

Among the children enrolled, 12.6% were HIV infected. Cameroon is said to be in a generalized HIV epidemic, although with decreasing prevalence, from 5.5 to 3.4% [20–22]. Pediatric HIV is also a major public health problem although transmission is reduced due to the

**Table 5. Occurrence of specific symptoms among actively dengue infected compared to non-infected patients.**

|  | IgM dengue |  |  |  |  |
| --- | --- | --- | --- | --- | --- |
|  | **No** | **Yes** | **Total** | **X2, p value** | **OR; 95% CI** |
| **Vomiting** | 32 (11.6%) | 12 (17.1%) | 44 (12.7%) | 1.577; 0.209 | 1.584; 0.769–3.263 |
| **Diarrhea** | 24 (8.7%) | 1 (1.4%) | 25 (7.2%) | 4.361; 0.036 | 0.153; 0.020–1.149 |
| **Fatigue** | 31 (11.2%) | 9 (12.9%) | 40 (11.5%) | 0.152; 0.697 | 1.171; 0.530–2.588 |
| **Total** | 277 | 70 | 347 |  |  |

implementation of option B+ (Treat all HIV infected pregnant or breastfeeding mother despite CD4 count or viral load) for the prevention of mother to child transmission [23]. The percentage of HIV obtained in this pediatric study population is in agreement with the national prevalence at the time of the study. Our study identified 21 out of 349 cases of dengue-HIV co-infections, thus a prevalence of 6%. Viral coinfections are known to be more severe in infants compared to adults. Viral coinfections could even be worse especially for dengue and HIV as the two viruses share the common target cells such as CD4 positive cells [19]. Special care should then be taken while consulting HIV infected children.

Because malaria and dengue are both transmitted by the bite of insects (Anopheles and Aedes respectively), environmental conditions favoring their breeding have an impact on their transmission rate. Stagnant water provide breeding conditions for Anopheles and Aedes. Parents should make sure their children sleep under protective bed nets which are freely distributed to pregnant women by the Malaria Control Programme in Cameroon, especially children younger than 5 years and during high transmission period. Our findings showed that children sleeping under bednet were less at risk from been infected by dengue and malaria, as bednet acts as shield and protects them from insect bites. The reason being that they are less at risk of been bitten by mosquitoes, although bednets are generally thought to have less impact on dengue, due to the day biting of dengue transmitting mosquito.

Malaria and HIV may be risk factors for dengue infections or vice-versa. Malaria and dengue are transmitted by mosquitoes that multiply rapidly during rainy seasons and thus may co-occur temporally. So being exposed to malaria can be concurrent to exposure to dengue, this is why the two infections can co-exist in the same person, and one being the risk factor for another. On the other side, HIV may be a risk factor for dengue and the reciproque may apply, given that both viruses target the same cells in the body.

## Conclusion

Dengue is common among Cameroonian children. Since Cameroon is endemic for malaria, fever may indicate either malaria or dengue. Clinicians should be aware of the presence of dengue in Cameroon and parents should properly use the bed net they are given to protect their children.

Management of patients with dengue fever and HIV infection should be studied, especially in infants where both viral diseases are sometimes more severe. Close monitoring may mitigate complications among HIV-infected individuals who contract dengue fever.

## Supporting information

**S1 Study Questionnaire.**
(PDF)

## Author Contributions

**Conceptualization:** Celine Nguefeu Nkenfou.

**Data curation:** Celine Nguefeu Nkenfou, Nadine Fainguem, Laeticia Grace Yatchou, Amidou Samie.

**Formal analysis:** Celine Nguefeu Nkenfou, Nadine Fainguem, Laeticia Grace Yatchou, Joel Josephine Kadji Kameni, Elise Lobe Elong, Amidou Samie, William Estrin.

**Investigation:** Celine Nguefeu Nkenfou, Félicitée Dongmo-Nguefack, Joel Josephine Kadji Kameni, William Estrin, Paul Ndombo Koki.

**Methodology:** Celine Nguefeu Nkenfou, Nadine Fainguem, Félicitée Dongmo-Nguefack, Laeticia Grace Yatchou, Joel Josephine Kadji Kameni, Elise Lobe Elong, Amidou Samie.

**Project administration:** Celine Nguefeu Nkenfou.

**Supervision:** Celine Nguefeu Nkenfou, William Estrin, Paul Ndombo Koki, Alexis Ndjolo.

**Validation:** Félicitée Dongmo-Nguefack, Amidou Samie, Paul Ndombo Koki, Alexis Ndjolo.

**Writing – original draft:** Celine Nguefeu Nkenfou.

**Writing – review & editing:** Celine Nguefeu Nkenfou, Nadine Fainguem, Félicitée Dongmo-Nguefack, Laeticia Grace Yatchou, Joel Josephine Kadji Kameni, Elise Lobe Elong, Amidou Samie, William Estrin, Paul Ndombo Koki, Alexis Ndjolo.

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
