## [Decision Letter · Decision Letter 0]

27 Feb 2020

Dear Dr. Nkenfou,

Thank you very much for submitting your manuscript "DENGUE INFECTION AMONG FEBRILE CHILDREN: PREVALENCE, CO-INFECTIONS AND ASSOCIATED FACTORS" for consideration at PLOS Neglected Tropical Diseases. As with all papers reviewed by the journal, your manuscript was reviewed by members of the editorial board and by several independent reviewers. In light of the reviews (below this email), we would like to invite the resubmission of a significantly-revised version that takes into account the reviewers' comments. 

Two reviewers have provided reviews for this paper. Both reviewers highlighted that this is an interesting study on an important area that presents novel findings. However they also pointed out that the manuscript needs substantial revisions. The reviewers have provided very thorough comments to help with this revision. I echo their comments and ask the authors to thoroughly revise the manuscript inline with these comments, but also with attention throughout to consistency and clarity of meaning of terms used, ensuring the focus of the paper is clear, with results presented clearly, and that the discussion leads from the results.

We cannot make any decision about publication until we have seen the revised manuscript and your response to the reviewers' comments. Your revised manuscript is also likely to be sent to reviewers for further evaluation.

Sincerely,

Hannah E Clapham

Guest Editor

Emily Gurley

Deputy Editor

Two reviewers have provided reviews for this paper. Both reviewers highlighted that this is an interesting study on an important area that presents novel findings. However they also pointed out that the manuscript needs substantial revisions. The reviewers have provided very thorough comments to help with this revision. I echo their comments and ask the authors to thoroughly revise the manuscript inline with these comments, with attention throughout to consistency and clarity of meaning of terms used, ensuring the focus of the paper is clear, with results presented clearly, and that the discussion leads from the results.

Reviewer's Responses to Questions

**Key Review Criteria Required for Acceptance?**

**Methods**

-Are the objectives of the study clearly articulated with a clear testable hypothesis stated?

-Is the study design appropriate to address the stated objectives?

-Is the population clearly described and appropriate for the hypothesis being tested?

-Is the sample size sufficient to ensure adequate power to address the hypothesis being tested?

-Were correct statistical analysis used to support conclusions?

-Are there concerns about ethical or regulatory requirements being met?

Reviewer #1: This manuscript describes a cohort of acutely febrile children presenting to a single center in Yaounde, Cameroon, with the objective of describing incidence of dengue virus (DENV) infection, and associated co-infection with malaria and/or HIV among children with undifferentiated febrile illness. The authors describe the various tests used to detect DENV, malaria, and HIV. For DENV, the authors describe using the SD Bioline dengue Duo, an IgM/IgG ELISA, and PCR. However, it is not clear how they defined a case of DENV. For example, in line 92, the authors state that a positive DENV NS1 test result prompted confirmatory testing by PCR. Were samples that were NS1 positive but PCR negative not considered cases? What about discrepancies between NS1 and IgM results? This confusion is reflected later in the results.

The authors' use of the term prevalence throughout the manuscript is confusing. In the DENV literature, as well as that of most other infections, seroprevalence refers to presence of serum antibody, typically IgG, describing evidence of previous exposure/infection. The authors report anti-DENV IgG results as signifying previous infection, but the use of the term prevalence to describe both previous exposure and incident infection is confusing. Prevalence and incidence are not interchangeable.

Reviewer #2: See below

**Results**

-Does the analysis presented match the analysis plan?

-Are the results clearly and completely presented?

-Are the figures (Tables, Images) of sufficient quality for clarity?

Reviewer #1: The presentation of the results is confusing. A diagram of how many subjects were classified into each of the diagnostic groupings would help to clarify the information provided in Table 2. Specifically, Table 2 indicates that there were 101 cases of DENV overall, with 70 cases positive for IgM. This implies that the remaining cases were due to NS1 positivity or PCR positivity. Yet, the text (lines 139-140) states that only 1 child was positive for NS1. A footnote in Table 3 explains that the 101 cases included both previous exposure and current DENV infections. The logic behind the grouping of previous exposure, signified by positive DENV IgG, together with incident infection, signified by positive DENV IgM, into a single group (referred to as "Ig dengue" in tables 3-5 but "Ag dengue" in tables 6-7?) is neither intuitive nor explained. DENV IgG in combination with positive IgM might signify non-primary infection with a serotype that is heterologous to that of the previous infection. However this is not discussed and such heterologous non-primary DENV infections are not identified.

As currently written, most of the results section simply restates information presented in the tables without providing additional interpretation. The tables can be presented in a more concise manner. There are seven tables, several of which (Tables 4 and 5 in particular) may be summarized in a sentence in the text or submitted as supplementary materials (see specific comments below). 

Since this manuscript is focused on DENV and associated co-infections, consider restructuring the results section to report DENV incidence first, then the co-infections in the context of DENV infection.

Reviewer #2: See below

**Conclusions**

-Are the conclusions supported by the data presented?

-Are the limitations of analysis clearly described?

-Do the authors discuss how these data can be helpful to advance our understanding of the topic under study?

-Is public health relevance addressed?

Reviewer #1: The authors appropriately highlight the public health relevance, that DENV infection occurs in children in Cameroon, frequently concurrently with either malaria and/or HIV, and that additional investigation of the effect of concurrent infection on disease presentation is warranted. However, discussion of associations between DENV and co-infection, which perhaps are the most interesting findings of the study, is superficial. Finally, the conclusion ends on speculation about disease severity, which is not addressed in this manuscript.

Reviewer #2: See below

**Editorial and Data Presentation Modifications?**

Reviewer #1: Specific comments:

1. Lines 54-55 are redundant

2. Line 60: There is a licensed vaccine (Dengvaxia, Sanofi-Pasteur)

3. Lines 86-87: During which year(s) was the study performed? Also, the statement regarding transmission during the rainy months requires a reference.

4. Consider providing the questionnaire in supplementary materials.

5. Line 92: Please specify a reference for the ELISA if this was an in-house assay, or the manufacturer if this was a commercial assay.

6. Line 99: Please clarify what is a "positive" urinary test? 

7. Line 115: Stating the upper age range as 180 months followed by the mean in years is awkward. Consider referring the upper limit as 15 years.

8. Line 117: Please clarify what "living in an environment with stagnant water" means.

9. Lines 128-129: 54 girls + 55 boys = 109 children. What about the other 74 children with malaria? Also Table 2 is referenced but does not include these data.

10. Table 2: Dengue (Overall cases) is reported as 101, while DENV co-infections were 15 + 68 + 21 = 104. All DENV cases were co-infections? Please clarify.

11. Table 2: Reporting of the CD4 count is poorly defined. Is this the mean or median? What does "Min:240-400" mean? Consider reporting this in the text instead of in the table, since the remainder of the table reports frequencies.

12. Lines 123-126 and table 2: Since this manuscript focuses on DENV, malaria, and HIV infection, consider reporting symptoms associated with these infections rather than the frequency of symptoms of the entire cohort.

13. Line 137: Is this the DENV case definition? What about the PCR testing referenced in the methods?

14. Table 4: Consider making this a supplementary table or excluding it altogether. The lack of association of any of the factors with sex can be concisely stated in the text. Additionally, the reference group should be identified when reporting odds ratios.

15. Table 5: Please clarify the reference group for odds ratios. Is it the odds of having a particular test result for the specified age group over the odds for all other ages? Or is there a reference group? Consider making this a supplementary table or excluding it altogether since the lack of association with age can be stated in the text.

16. Lines 206-207: Please clarify the statement regarding susceptibility to infection and mother-to-child transmission. 

17. Line 224: The principal targets of DENV infection are monocytes/macrophages/dendritic cells, whereas the principal targets of HIV infection are T lymphocytes. These are not the same cells.

18. Line 231: Malaria and DENV are transmitted by mosquitoes, not flies.

19. Lines 232-234: These statements are unsupported. DENV and malaria are transmitted by different mosquito vectors with different feeding habits (daytime vs. night time). Thus exposure is inherently different.

Reviewer #2: See below

**Summary and General Comments**

Reviewer #1: This manuscript reports the incidence of dengue virus (DENV) infection among children in Cameroon, along with associated co-infection with malaria and HIV. Given the paucity of data on DENV in Cameroon, the data offer an important contribution toward advancing knowledge of DENV in Africa. Unfortunately, this manuscript is disorganized and presents the data in a confusing fashion. The analyses are superficial and/or incomplete. These shortcomings detract from the potential impact of the study.

Reviewer #2: The authors describe results of an enhanced passive surveillance study wherein children aged 6 months to 15 years presenting with undifferentiated fever were enrolled and tested simultaneously (I believe) for malaria, dengue, and HIV. The authors note that co-infections may be missed using standard testing practices. The results are quite interesting and could represent an important contribution to the literature regarding the clinical overlap between dengue, malaria, and HIV. However, the manuscript would benefit from increased precision in presenting results and discerning current and prior dengue infection based upon serological results. 

MAJOR COMMENTS

- Please place the study type in the abstract and title (presumably enhanced passive surveillance?). Provide further details about the methods and very basic descriptive statistics (number enrolled, age, etc,) in abstract. 

- I don’t know that it is reasonable to conclude that DENV IgG-positive alone represents ‘dengue-infected’ – they may have had historical infection. I do agree that DENV IgM is more likely to be DENV (but may also represent ZIKV or some other cross-reactive flavivirus). I also agree that NS1-positive for DENV = acute infection. Similarly, for ‘co-infections’ I would not report that malaria smear + / DENV IgG + represents a co-infection. 

- Please be more precise in your presentation of findings – for example, in discussion line 209 – our findings showed a prevalence of 52.7% of malaria cases. You found that 52.7% of children presenting for care with acute undifferentiated fever had evidence of malaria sporozoites (correct?). 

- Please comment on how this testing algorithm differs from common practice in Cameroon (I’m guessing that malaria testing would be the first step, stop if positive?) and how your findings may indicate that an expanded approach may be needed. 

- Relatedly, please comment on ‘next steps’ needed to better understand the clinical presentation and epidemiology of dengue and malaria / HIV coinfections in Cameroon. Your findings are very interesting and suggest that there may be opportunities to better understand how clinical severity and complications of dengue infection may be changed in the setting of co-infection. 

MINOR COMMENTS

- Please add ‘Cameroon’ and the general study type to the full title

- Introduction.

o Line 60. There is in fact a licensed vaccine for dengue available in many countries (though not, to my knowledge, in Cameroon), however there are restrictions on the use given safety concerns. 

- Has there been an entomological survery done in Cameroon to document the geographic range of Aedes aegypti and Aedes albopictus? If so, would be good to mention. If not, would also be good to mention this knowledge gap. 

- Would refer to DENV ‘serotypes ’ 1 and 2

- Please clarify that HIV, malaria, and dengue testing was performed for all individuals?

- Was there a time limit for the duration of fever? (I.e., less than 10 days?)

- Did you ask if the children slept under a bednet during the day? This is the biting period for Aedes mosquitoes

- For table 1, the percents for stagnant water don’t add up to 2100%

- Table 2: CD4 counts – is this the median?

- Results, would change the ORs so that they reflect your description of results (e.g., line 155, ‘were less at risk of being infected’ – change OR to be (1/8.85)

- Tables 6 and 7 seem to be mixed up in the text

- Tables 6-7 – does ‘Ag dengue’ refer to ‘any immunoglobulin, IgM or IgG, to DENV’?

- Table 6 – Does VIH refer to HIV?

- For all tables, footnotes defining abbreviations and the relevant diagnostic tests would be valuable

- It is unclear that is meant in line 206 about ‘findings are supported by reports showing more susceptibility to infection and cases of mother to child transmission of dengue virus’

PLOS authors have the option to publish the peer review history of their article (what does this mean?). If published, this will include your full peer review and any attached files.

Reviewer #1: No

Reviewer #2: No
---

## [Decision Letter · Decision Letter 1]

9 Sep 2020

Dear Dr. Nkenfou,

Thank you very much for submitting your manuscript "ENHANCED PASSIVE SURVEILLANCE OF DENGUE INFECTION AMONG FEBRILE CHILDREN: PREVALENCE, CO-INFECTIONS AND ASSOCIATED FACTORS IN CAMEROON" for consideration at PLOS Neglected Tropical Diseases. As with all papers reviewed by the journal, your manuscript was reviewed by members of the editorial board and by several independent reviewers. In light of the reviews (below this email), we would like to invite the resubmission of a significantly-revised version that takes into account the reviewers' comments. 

We cannot make any decision about publication until we have seen the revised manuscript and your response to the reviewers' comments. Your revised manuscript is also likely to be sent to reviewers for further evaluation.

Sincerely,

Hannah E Clapham

Associate Editor

Emily Gurley

Deputy Editor

In addition to Reviewer 2 comments below, the AE has the following additional comments to help with the editing as suggested by Reviewer 2. This paper details an important study, but the analysis and the description of the results requires substantial editing. 

Abstract: 

Severe disease is generally refered to as severe dengue at the moment: https://www.who.int/news-room/fact-sheets/detail/dengue-and-severe-dengue. 

“Dengue overall seroprevalence was 29.1%. Malaria cases were 52.7% and HIV cases represent 12.6%.” this line is unclear suggest rephrasing, seroprevalence suggests past infection, where as malaria and HIV are refered to as cases- please clarify what was being assessed.

“But malaria either due to P vivax or P

36 falciparum was associated to dengue infection, P=0.001 and 0.003 respectively. Likewise, HIV was

37 significantly associated to dengue p=0.028.” It is unclear what is being assessed here- suggest rephrasing. 

Author summary: 

“Dengue

45 fever, transmitted by a mosquitoe as well as malaria is common among Cameroonian children.” Suggest rephrasing. 

Introductions: 

“Both infections share similar clinical symptoms” requires reference.

There is now a vaccine for dengue- https://www.who.int/immunization/diseases/dengue/revised_SAGE_recommendations_dengue_vaccines_apr2018/en/

Results: 

P9 Line 137: Rephrase to a complete sentence. 

Similar points throughout results, please ensure all sentences are complete sentences. 

What are the (%) in Table 3? This is currently very unclear, as is the interpretation from this table for when the peak transmission occurred. Much clarification needed. 

What are the (%) in Tables 5 and 6? This requires clarification/editing. 

LIne 182: I think the reference to table 6 and 7 are reversed. In addition, please provide further description of the co-infection relationship. At the moment in the text it is hard to understand what is meant. 

I agree with reviewer 2, there needs to be further clarity on the past dengue infections and current dengue infections definition. In addition, how was a co-infection with dengue and the other pathogens defined? I think this should be current dengue infection, but it is not clear if this was how it was defined. This needs additional care and clarity for dengue, due to the acute nature of the infection and that the test used measure IgG and IgM (and clarity is needed on the interpretation of IgG positivity in this assay). Table 2 and the discussion of the results in the text needs to be clearer.

Similarly, in the discussion what is meant by “prevalence” is this the % of febrile cases that are current dengue, or past dengue? I don’t think either is the “prevalence”. Please rephrase. 

“Our findings are supported by reports showing more susceptibility to

207 infection and cases of mother to child transmission of dengue virus 16-17.” Please be clear which of your results are supported by this. 

The dengue- HIV co-infection discussion requires clarification. 

“the common target cells”. Please state the target cells, and provide references.

Reviewer's Responses to Questions

**Key Review Criteria Required for Acceptance?**

**Methods**

-Are the objectives of the study clearly articulated with a clear testable hypothesis stated?

-Is the study design appropriate to address the stated objectives?

-Is the population clearly described and appropriate for the hypothesis being tested?

-Is the sample size sufficient to ensure adequate power to address the hypothesis being tested?

-Were correct statistical analysis used to support conclusions?

-Are there concerns about ethical or regulatory requirements being met?

Reviewer #2: See below

**Results**

-Does the analysis presented match the analysis plan?

-Are the results clearly and completely presented?

-Are the figures (Tables, Images) of sufficient quality for clarity?

Reviewer #2: See below

**Conclusions**

-Are the conclusions supported by the data presented?

-Are the limitations of analysis clearly described?

-Do the authors discuss how these data can be helpful to advance our understanding of the topic under study?

-Is public health relevance addressed?

Reviewer #2: See below

**Editorial and Data Presentation Modifications?**

Reviewer #2: See below

**Summary and General Comments**

Reviewer #2: The manuscript has been improved from the prior submission; the authors’ thoughtful revisions are much appreciated. The discussion, in particular, is more solid. The classification of dengue illnesses as past and current remains a confusing aspect of the manuscript, and multiple small typological errors / areas for further clarification remain. 

General comments:

- Not necessary to separate out IgM+ / IgG- and IgM+ / IgG+, this makes the presentation of results confusing. The timeline of specimen collection relative to onset of symptoms may influence whether IgM alone, IgM/IgG or IgG alone were detected but it is safe to say that detectable IgM+ is suggestive of acute / recent DENV infection. IgG+ alone suggests prior exposure. This simplification of presentation alone would clarify the abstract, results, table 2, and discussion. 

- Relatedly, please comment on the timing of specimen collection relative to the onset of symptoms (median and IQR). Perhaps add to table 1. 

- The manuscript would benefit from a detailed review to correct typological errors and general flow of language. This reviewer has pointed out a few specific areas to improve the flow of language below, but multiple other small typos remain. (e.g., abstract, line 40 ‘pic’ should be ‘peak’; Line 137, ‘with minimum been 1 day’ should be ‘with a minimum of 1 day’)

Minor comments:

- It is standard convention to define dengue virus as DENV, then use it throughout to refer to serotypes. ‘dengue’ is fine to use when describing dengue illnesses. 

- The use of the term ‘prevalence’ is applied to very distinct concepts, i.e., % of febrile illnesses identified as dengue, seroprevalence. For percent of illness cases identified as dengue, perhaps move away from prevalence and describe as ‘the proportion of febrile illnesses identified as dengue’, for example. 

- Line 70. Dengvaxia may prevent some infections but the major contribution appears to be a reduction in hospitalized illness, albeit with an increase in the incidence of severe disease among young, dengue -naïve children. 

- Line 136 and Table 1. Water flasks or stagnant water? Does water flasks mean ‘containers’?

- Lines 142-146. For each of these infections, are symptoms reported for single infections (i.e., HIV alone, no dengue or malaria) or for a given infection regardless of coinfections? Consider a supplementary table showing symptoms for each virus alone, coinfections, comparing severity and clinical presentation. 

- Lines 166-167 is very confusing. Hopefully can be simplified as in first general comment above. 

- Table 2. Please be explicit in table about the fact that you are considering acute dengue (not prior dengue) in coinfection estimates (as you have added to the text).

- Table 3 – perhaps turn into a figure showing percents by month?

- Supplemental table 4. Please indicate what is the referent group for the odds ratios (i.e., are the odds of IgG lower in females?)

- Supplemental table 4. Please indicate what is the referent group for age the odds ratios i.e., is it age 2-5 years compared to all other age groups? You may consider making 0-2 the referent group for subsequent age groups)

- Line 290. This sentence about viral infections being more severe in children could be softened. Chickenpox, for example, is more severe in adults.

PLOS authors have the option to publish the peer review history of their article (what does this mean?). If published, this will include your full peer review and any attached files.

Reviewer #2: No
---

## [Decision Letter · Decision Letter 2]

13 Dec 2020

Dear Dr. Nkenfou,

Thank you very much for submitting your manuscript "ENHANCED PASSIVE SURVEILLANCE OF DENGUE INFECTION AMONG FEBRILE CHILDREN: PREVALENCE, CO-INFECTIONS AND ASSOCIATED FACTORS IN CAMEROON" for consideration at PLOS Neglected Tropical Diseases. As with all papers reviewed by the journal, your manuscript was reviewed by members of the editorial board and by several independent reviewers. In light of the reviews (below this email), we would like to invite the resubmission of a significantly-revised version that takes into account the reviewers' comments. Reviewers continue to believe that the topic of your paper is relevant and of interest to our audience, but this revision did not satisfactorily address comments noted on the first submission.

We cannot make any decision about publication until we have seen the revised manuscript and your response to the reviewers' comments. Your revised manuscript is also likely to be sent to reviewers for further evaluation.

Sincerely,

Hannah E Clapham

Associate Editor

Emily Gurley

Deputy Editor

Reviewer's Responses to Questions

**Key Review Criteria Required for Acceptance?**

**Methods**

-Are the objectives of the study clearly articulated with a clear testable hypothesis stated?

-Is the study design appropriate to address the stated objectives?

-Is the population clearly described and appropriate for the hypothesis being tested?

-Is the sample size sufficient to ensure adequate power to address the hypothesis being tested?

-Were correct statistical analysis used to support conclusions?

-Are there concerns about ethical or regulatory requirements being met?

Reviewer #2: See below

**Results**

-Does the analysis presented match the analysis plan?

-Are the results clearly and completely presented?

-Are the figures (Tables, Images) of sufficient quality for clarity?

Reviewer #2: See below

**Conclusions**

-Are the conclusions supported by the data presented?

-Are the limitations of analysis clearly described?

-Do the authors discuss how these data can be helpful to advance our understanding of the topic under study?

-Is public health relevance addressed?

Reviewer #2: See below

**Editorial and Data Presentation Modifications?**

Reviewer #2: See below

**Summary and General Comments**

Reviewer #2: The authors have made some significant improvements in the precision with which results are presented compared to the previous drafts. However, significant ambiguity and inconsistency remain with the inter-mixing of how dengue ‘cases’ are defined (for example, how IgM and IgG results are interpreted). Further, numerous typological / grammatical errors remain. 

MAJOR COMMENTS.

- There remains some mixing of definitions of a dengue ‘case’ or dengue-infected individual. For example, in Table 2, a case seems to be defined as IgG or IgM (+). This is in contradiction to the primary aim of the study, which is to define ‘the prevalence’ (incidence?) of dengue in a cohort of children presenting with fever. IgG would reflect past infection (seroprevalence, not necessarily active infection) while IgM would reflect current or recent infection. Table 6, IgG antibodies are not necessarily suggestive dengue infection (as in the table heading) and it is confusing to consider them as such in the table. For the purposes of the study, would define a dengue ‘case’ as IgM only and use these definitions more consistently. 

- Relatedly, please be consistent with your definition of ‘seroprevalence’ – Discussion, lines 196-197 – authors use seroprevalence to refer to both IgM and IgG. This is confusing and, again, reflects a mixing of definitions for acute and historical infection. 

- The authors may switch to using terms such as ‘the proportion of children presenting with fever who were found to have acute dengue infection was XX’ rather than ‘the prevalence of dengue’ – this is an acute infection (in contrast to malaria) and the over-reliance on the term throughout the manuscript to describe acute infections in the study is confusing when inter-mixed with references to seroprevalence.

INTRODUCTION:

- Lines 53-54: recommend shortening to “dengue…is the most prevalent mosquito-borne viral disease worldwide and is described as a ‘neglected’….”

- Line 59-60. There is a licensed dengue vaccine available in many countries, but safety concerns and screening requirements limit its widespread use and impact

- Line 65-66: A study of prevalence and distribution – looking at what population? Adults / children / all presenting with fever? This influences interpretation. 

- Line 66: A study conducted in 2009 – what kind of study?

- Line 68: What type of climate? (Tropical? Subtropical?)

- Line 72. Two serotypes identified (DENV-1 and DENV-2)

- Lines 75-76 – would make very clear: the objective of the study was to determine…and associated risk factors in a pediatric population presenting with fever in Yaounde, Cameroon.

METHODS:

- Years of the study? Duration of the study? April to October of one year?

- Any limits to the duration of fever for inclusion criteria? (i.e., within 7 days of onset of fever?)

- Lines 102-103: Please refine sentence such as “Infants were enrolled only when their parent or guardian gave written proxy-consent” – but was this true only for infants? Assent obtained from older minors?

- Lines 134-138. Would move these definitions to the methods. Please rephrase “Were considered exposed” and “Were considered dengue infected” to be something like “Therefore, current or recent dengue infection was defined on the basis of a positive result for NS1 Ag and / or IgM.”

RESULTS:

- Clinical characterization: please provide information on the duration of fever, if known

- Line 123: “Highest temperature was 41”, “Average temperature was…” 

- Line 123: Did you collect data only on active symptoms, or history of symptoms? For example, “12.6% reported a history of vomiting” versus “12.6% reported current symptoms of vomiting at the time of enrollment”

- Line 141.” The average CD4 count was 969 cells/mm3” “Children have a lower CD4 count were more likely to be positive”(Statistical test used, p value?)

- Line 144. “The rate of Malaria – dengue coinfection was 19.5%”

- Line 145. “4.3% of the study population had evidence of current or recent dengue infection, as well as both malaria and HIV” (is this correct?

- Line 152. What does “small raining season mean” – it was a low year for rainfall?

- Table 4. If the lowest age group is referent, why is the OR not 1?

- Table 5. What is referent? Presumably, the absence of the other infection?

- Table 6. Presence of dengue IgG antibody is not consistent with your definition of ‘dengue infection’ rather ‘prior dengue exposure’. Please consider removing this comparison from the table – why would we think that historical exposure to dengue would be associated with diarrhea, for example?

- Lines 178-179. As above. 

DISCUSSION: 

- Lines 196-197. Confusing to use seroprevalence to refer to both IgM and IgG. 

- Line 205. Please comment specifically on how your methodology may underestimate coinfection compared to the prior study. 

- Line 214. Recommend you remove the sentence beginning “In such setting”

- Line 216. “Viral coinfections”

- Line 219. Capitalize Aedes and Anopheles

- Line 224-225. Aedes and Anopheles don’t exactly breed in the same conditions, but both may be increased during rainy seasons and thus may co-occur temporally

- Line 226-227. Please refine final sentence, it is a little confusing

- Please comment on the finding that sleeping under a bednet or living in an area without stagnant water was protective against dengue. These aren’t classically thought of as protective against Aedes – is this reflective of some other, related risk factor? (Socioeconomic status, etc?)

- You may consider commenting that the seroprevalence increased steadily with age (in fact, this may be a good figure to include, perhaps combining tables 3 and 4 to make space for a new figure) – this suggests that transmission is somewhat high and endemic (rather than epidemic, where we may be more even distribution across ages)

PLOS authors have the option to publish the peer review history of their article (what does this mean?). If published, this will include your full peer review and any attached files.

Reviewer #2: No
---

## [Editor Report · Decision Letter 3]

23 Feb 2021

Dear Dr. Nkenfou,

Thank you very much for submitting your manuscript "ENHANCED PASSIVE SURVEILLANCE OF DENGUE INFECTION AMONG FEBRILE CHILDREN: PREVALENCE, CO-INFECTIONS AND ASSOCIATED FACTORS IN CAMEROON" for consideration at PLOS Neglected Tropical Diseases. As with all papers reviewed by the journal, your manuscript was reviewed by members of the editorial board and by several independent reviewers. The reviewers appreciated the attention to an important topic. Based on the reviews, we are likely to accept this manuscript for publication, providing that you modify the manuscript according to the review recommendations. 

We thank the authors for their revisions. 

There remain some sections of the manuscript that are still in need of refinement. Please see the comments at the end of this email message. 

Sincerely,

Hannah E Clapham

Associate Editor

Emily Gurley

Deputy Editor

We thank the authors for their revisions. 

There remain some sections of the manuscript that are still in need of refinement. Please see the comments below. 

"Past" not "passed"

The following is still not a full sentence, please edit to a full sentence, for example “The highest temperature observed was…etc”: Highest temperature was 41°C, lowest temperature was 37.8°C and the average temperature was 38.9°C.

“Children having the lower CD4 count (240-400 CD4 cells/mm3) were more likely to be positve for malaria (20% compared to 10.8%; X2=0.71, p=0.4), HIV (40% compared to 0%; X2=31.08; p<0.001) or dengue (20% compared to 5.3%; X2 = 2.96, p=0.085) .” This only appears to be positive to HIV, so I would suggest rephrasing these results as it currently reads like you are suggesting that there are relationships with all HIV, dengue and malaria. 

“the reason being that they are less at risk of been bitten by flies”. This should be in the discussion, and the word flies needs to be replaced with either mosquito or vector. Similar for later reference to “flies”.

“Stagnant water provide breeding conditions for Anopheles and Aedes.” This also should be in the discussion. 

“IgG represents cases of previous exposure to dengue” I would suggest removing “cases of” here for clarity. 

Discussion:

Please add to the comment about bednets being protective against dengue too, as bednets are generally thought to have less impact on dengue, due to the day biting of dengue transmitting mosquito. As the review suggested, could this also be a soci-economic factor or something else here? 

“Among the children enrolled, 12.6% were HIV infected. Cameroon is said to be in a generalized HIV epidemic, although with decreasing prevalence, from 5.5 to 3.4% 21-23. Pediatric HIV is also a major public health problem although transmission is reduced due to the implementation of option B+ (Treat all HIV infected pregnant or breastfeeding mother despite CD4 count or viral load) for the prevention of 235 mother to child transmission 24. The percentage of HIV obtained in this study is in agreement with the national prevalence at the time of the study. Our study identified 21 out of 349 cases of dengue-HIV co infections, thus a prevalence of 6%.” I don’t understand the two % for HIV prevalence here- please rephrase. 

“The result obtained in the present study is higher than the 9.8% previously reported by Demanou et al in Yaoundé among adults 12.” Please give more details on this study and which of the previous reported results from your study this is being compared to.

Figure Files:

Data Requirements:

Reproducibility:

References

---

## [Editor Report · Decision Letter 4]

16 Mar 2021

Dear Dr. Nkenfou,

We are pleased to inform you that your manuscript 'ENHANCED PASSIVE SURVEILLANCE OF DENGUE INFECTION AMONG FEBRILE CHILDREN: PREVALENCE, CO-INFECTIONS AND ASSOCIATED FACTORS IN CAMEROON' has been provisionally accepted for publication in PLOS Neglected Tropical Diseases.

Best regards,

Hannah E Clapham

Associate Editor

Emily Gurley

Deputy Editor

---

## [Editor Report · Acceptance letter]

3 Apr 2021

Dear Dr. Nkenfou,

We are delighted to inform you that your manuscript, "ENHANCED PASSIVE SURVEILLANCE OF DENGUE INFECTION AMONG FEBRILE CHILDREN: PREVALENCE, CO-INFECTIONS AND ASSOCIATED FACTORS IN CAMEROON," has been formally accepted for publication in PLOS Neglected Tropical Diseases.

Best regards,

Shaden Kamhawi

co-Editor-in-Chief

Paul Brindley

co-Editor-in-Chief
